# Risk of prostate cancer in relatives of prostate cancer patients in Sweden: A nationwide cohort study

Xing Xu[1,2☯], Elham Kharazmi[1,3,4☯], Yu Tian[1,2], Trasias Mukama[1,2], Kristina Sundquist[3,5,6], Jan Sundquist[3,5,6], Hermann Brenner[1,7,8], Mahdi Fallah[1,3]*

1 Division of Preventive Oncology, National Center for Tumor Diseases (NCT), German Cancer Research Center (DKFZ), Heidelberg, Germany, 2 Medical Faculty Heidelberg, Heidelberg University, Heidelberg, Germany, 3 Center for Primary Health Care Research, Lund University, Malmö, Sweden, 4 Institute of Medical Biometry and Informatics, Heidelberg University Hospital, Heidelberg, Germany, 5 Department of Family Medicine and Community Health, Department of Population Health Science and Policy, Icahn School of Medicine at Mount Sinai, New York City, New York, United States of America, 6 Center for Community-based Healthcare Research and Education (CoHRE), Department of Functional Pathology, School of Medicine, Shimane University, Izumo, Japan, 7 Division of Clinical Epidemiology and Aging Research, German Cancer Research Center (DKFZ), Heidelberg, Germany, 8 German Cancer Consortium (DKTK), German Cancer Research Center (DKFZ), Heidelberg, Germany

☯ These authors contributed equally to this work.
* m.fallah@dkfz.de

**Data Availability Statement:** This study made use of the Swedish Cancer Registry data. Data from the nationwide registers cannot be shared by study authors, however further information and relevant

## Abstract

### Background

Evidence-based guidance for starting ages of screening for first-degree relatives (FDRs) of patients with prostate cancer (PCa) to prevent stage III/IV or fatal PCa is lacking in current PCa screening guidelines. We aimed to provide evidence for risk-adapted starting age of screening for relatives of patients with PCa.

### Methods and findings

In this register-based nationwide cohort study, all men (aged 0 to 96 years at baseline) residing in Sweden who were born after 1931 along with their fathers were included. During the follow-up (1958 to 2015) of 6,343,727 men, 88,999 were diagnosed with stage III/IV PCa or died of PCa. The outcomes were defined as the diagnosis of stage III/IV PCa or death due to PCa, stratified by age at diagnosis. Using 10-year cumulative risk curves, we calculated risk-adapted starting ages of screening for men with different constellations of family history of PCa. The 10-year cumulative risk of stage III/IV or fatal PCa in men at age 50 in the general population (a common recommended starting age of screening) was 0.2%. Men with ≥2 FDRs diagnosed with PCa reached this screening level at age 41 (95% confidence interval (CI): 39 to 44), i.e., 9 years earlier, when the youngest one was diagnosed before age 60; at age 43 (41 to 47), i.e., 7 years earlier, when ≥2 FDRs were diagnosed after age 59, which was similar to that of men with 1 FDR diagnosed before age 60 (41 to 45); and at age 45 (44 to 46), when 1 FDR was diagnosed at age 60 to 69 and 47 (46 to 47), when 1 FDR was diagnosed after age 69. We also calculated risk-adapted starting ages for other

contact details can be found on: https://www.
socialstyrelsen.se/en/statistics-and-data/registers/
register-information/ Postal address:
Socialstyrelsen, SE-106 30 Stockholm, Sweden
Phone: +46 (0)75 247 30 00 Fax: +46 (0)75 247 32
52 E-mail: socialstyrelsen@socialstyrelsen.se Links
for Swedish Cancer Registry: https://www.
socialstyrelsen.se/en/statistics-and-data/registers/
register-information/swedish-cancer-register/
Email address: cancerregistret@socialstyrelsen.se.

**Funding:** XX received scholarship from the China
Scholarship Council (www.
chinesescholarshipcouncil.com/). KS (grant
number 2018-02400) and JS (grant number 2020-
01175) were supported by grants from the
Swedish Research Council (www.vr.se/english.
html). The funders had no role in study design,
data collection and analysis, decision to publish, or
preparation of the manuscript.

**Competing interests:** The authors have declared
that no competing interests exist.

**Abbreviations:** ACS, American Cancer Society;
AUA, American Urological Association; CI,
confidence interval; FDR, first-degree relative; ICD-
7, International Classification of Diseases-7th
Revision; PCa, prostate cancer; PSA, prostate-
specific antigen; RECORD, REporting of studies
Conducted using Observational Routinely-collected
health Data; TNM, Tumor, Nodes, Metastases;
USPSTF, United States Preventive Services Task
Force.

benchmark screening ages, such as 45, 55, and 60 years, and compared our findings with those in the guidelines. Study limitations include the lack of genetic data, information on life-style, and external validation.

## Conclusions

Our study provides practical information for risk-tailored starting ages of PCa screening based on nationwide cancer data with valid genealogical information. Our clinically relevant findings could be used for evidence-based personalized PCa screening guidance and supplement current PCa screening guidelines for relatives of patients with PCa.

## Author summary

### Why was this study done?

- Family history is the strongest known risk factor for prostate cancer (PCa), and current guidelines concur that an earlier screening for men with a family history of PCa is necessary.

- However, limited evidence-based guidance is available on at what age actually this early screening should start.

- This study was conducted to provide precise recommendations about at what age should relatives of PCa patients start screening based on the number of affected relatives and the age at onset of PCa in the family.

### What did the researchers do and find?

- In this nationwide study on 6,343,727 men, the risk of stage III/IV or fatal PCa in close family members of patients with PCa was estimated.

- It was observed that men with family history of PCa reach the screening risk threshold up to 12 years earlier than the general population.

- This study found that age, age at diagnosis of PCa in relative/s, and number of affected first-degree relatives (FDRs) are important elements in increased risk of stage III/IV or fatal PCa, and these factors accordingly resulted in different risk-adapted starting ages of PCa screening.

- Comparison between our evidence-based risk-adapted starting age of screening and recommended age of PCa screening by different guidelines showed a difference ranging from −2 to 11 years.

### What do these findings mean?

- This study made use of the largest dataset available, to our knowledge, to identify the optimal age for starting PCa screening in relatives of patients with PCa.

- This study took into account not only the number of relatives but also age at onset of PCa in the family members, which is an additional important piece of information for the guidelines.

- The results may contribute to a more evidence-based personalized PCa screening guidance in real-world settings, and clinicians could inform patients with PCa about this possibility and encourage individualized counseling for their relatives.

## Introduction

Prostate cancer (PCa) is the second most common cancer in men (age-standardized incidence rate 29.3 per 100,000) and a major cause of cancer deaths (age-standardized mortality rate 7.6 per 100,000) in the world [1]. The etiology of PCa has remained poorly understood, and measures for primary prevention are scarce. Age, ethnicity, and family history are known risk factors for PCa [2].

Because of growing concerns about overdiagnosis and overtreatment of indolent and early-stage PCa, the United States Preventive Services Task Force (USPSTF) recommended against the routine use of the prostate-specific antigen (PSA) testing for PCa screening [3–7]. Thereafter, continuous decline in the overall incidence of PCa has been observed. However, the incidence of late-stage PCa with poor prognosis has increased [6]. The Task Force revised their recommendation in 2018 for men aged 55 to 69 years to an informed decision-making about the PSA testing for PCa screening when an updated review suggested that screening offers a benefit in reducing PCa mortality in this age group [8,9]. However, they were not able to make a separate, specific recommendation on PSA-based screening for PCa in men with a family history of PCa, due to lack of evidence-based information in this regard [9].

It is generally accepted that men with a family history of PCa are at higher risk of developing this cancer and should be considered as high-risk group for PCa. Previous studies have reported that having a brother or father with PCa doubles a man's risk of developing this cancer [10,11]. Earlier screening in family members of patients with PCa as high-risk population has been suggested by some guidelines and recommendations [9,12–18]. However, there is concern about overdiagnosis of indolent and very early-stage PCa in relatives of patients with PCa, which may result in overtreatment and more harm than benefit for them [8,19,20]. Limited evidence-based guidance is available on exactly at what age should screening in relatives of patients with PCa be started. In other words, current guidelines in this regard can be perceived as an extrapolated and rather intuitive best guess concerning the starting age of screening. Guidelines for screening in such affected families lack precise recommendations according to the number of affected relatives and the onset of PCa in the family. Therefore, further investigations are needed to provide the risk of life-threatening PCa (late-stage or fatal PCa) in family members of patients with PCa who may benefit most from earlier screening and treatments possibilities. Furthermore, the question of "At what age these high-risk men should undergo screening?" warrants an evidence-based answer. To fill in these gaps, we aimed to provide clinically relevant evidence to be used for risk-adapted starting age of screening for relatives of patients with PCa who are at high risk of stage III/IV or fatal PCa using the world's largest nationwide register-based family-cancer datasets.

## Methods

All men residing in Sweden who were born between 1932 and 2015 along with their fathers were included in this study. Information from the Multi-Generation Register, national censuses, Swedish Cancer Registry, and death notifications were linked using a unique national identification number. These datasets have been described in detail elsewhere [21]. In brief, data on family relationships were obtained from the Multi-Generation Register, where children born after 1931 are registered with their parents as families. In this nationwide cohort study, full information on participants' fathers' life spans (if not restricted by beginning/end of cancer registry data or immigration/emigration) was available in the study, but offspring were followed up to age 84 years. This register was linked to the Cancer Registry Data from 1958 to 2015. A 4-digit diagnostic code according to the International Classification of Diseases-7th Revision (ICD-7) and subsequent ICD classifications were available. The underlying cause of death was available from the Swedish Cause of Death Register. The Swedish Cancer Registry and Cause of Death Register have been reported to have high level of completeness (about 96%) and accuracy [22,23]. The 2015 version of the database (updated in 2017) includes >12.8 million individuals (6,343,727 men) with available genealogical information, and >2.2 million primary invasive cancer records.

The Swedish National Board of Health and Welfare does not recommend a countrywide population-based PSA screening program for men [24]. However, to mitigate the overestimation of familial risk due to aggregation of indolent (opportunistic screen-detected) cancers in the families, we used the occurrence of stage III/IV or fatal PCa as the outcome. The exposure of interest was having a history of PCa (regardless of Tumor, Nodes, Metastases [TNM] stage or cancer-specific death status) in first-degree relatives (FDRs: father, brother, or son). The outcome was either diagnosis of stage III/IV PCa at the time of first diagnosis (based on the American Joint Committee on Cancer, 8th edition of cancer TNM staging) and/or death due to PCa [25]. The death was considered due to PCa when PCa was reported as the main underlying cause of death. As data on PSA and Gleason score/grade group were not available in the Swedish Cancer Registry, in this study, only those with "T3-4, N0, and M0" status were considered as stage III PCa. Those with "T1-2, N0, and M0" status were not considered as "with outcome" unless they died of PCa. These mean that the cancer has grown outside the prostate, broken through the capsule (covering) of the prostate gland, and might have spread to the seminal vesicles (T3), or it has spread into other tissues next to the prostate, such as the urethral sphincter, bladder, rectum, and/or the wall of the pelvis (T4). It has not spread to nearby lymph nodes (N0) or elsewhere in the body (M0). Stage IV was defined as either "Any T, N1, and M0" or "Any T, Any N, and M1," which means that the cancer has spread to either nearby lymph nodes (N1) or elsewhere in the body (M1). The follow-up started at birth, immigration date, or starting date of the study, January 1, 1958, whichever came latest. The follow-up ended on the year of PCa diagnosis, year of death, emigration, or closing date of the study, December 31, 2015, whichever came earliest.

In this study, a dynamic definition of family history of PCa (and not the static approach) was used to consider the changes in family history over time. Both methods and the superiority of the dynamic approach for the purpose of our risk stratification study have been discussed in detail elsewhere [26]. In brief, the dynamic family history of PCa in every participant was defined at entry into the study, and it changed every time that a new family member was diagnosed with PCa. For instance, if there was an index man with 2 brothers who were diagnosed with PCa in 1992 and 1998 and he himself was diagnosed with PCa in 1995, under the dynamic definition of family history, the brother diagnosed in 1998 would not be counted into the family history of index man since in real-world scenario, one does not know the future history of

their relatives. When a man had only 1 FDR diagnosed with PCa, he would be considered as men in group "1 FDR." Once the second FDR was diagnosed, he would shift to the group ">1 FDR" until his own diagnosis of stage III/IV or fatal PCa or the end of follow-up. This represents the actual change in the real-world when men mostly learn about the diagnosis of their close relatives at different time points of their life. For each year in the follow-up of individuals, we updated the family history profile for every index man, representing the real-time status (number and youngest age at diagnosis among relatives) of individuals with PCa diagnosed in his family. Occurrence of PCa in family members after the date of PCa diagnosis in the index patient was not considered in familial risk calculations.

The 10-year cumulative risk was calculated based on the following equations [27]:

- *Age-specific annual incidence rate = Number of cases during each 1-year follow-up divided by person-years*

- *10-year cumulative rate for age X = Sum of 10 consecutive annual age-specific incidence rates from age X to age X+9*

- *10-year cumulative risk = 1 –exp $^{(-10\text{-year cumulative rate})}$*

Exact values for person-years from individual data were used in the calculation of cumulative incidences. Risk-adapted starting ages of screening were determined by the age at which men with different constellations of family history of PCa reached the same level of risk as men in the general population at benchmark ages 40, 45, 50, 55, or 60 years. This approach of calculating risk-adapted starting age of cancers has already been already applied to some other cancers [28–30]. The 95% confidence intervals (CIs) of age-specific 10-year cumulative risk were calculated using the 2.5 and 97.5 percentiles of the bootstrap estimate distribution made by bootstrapping method (200 replications), from which CIs of proposed starting ages (ages of reaching mass screening level of risk) were derived according to the abovementioned formula for conversion of rate to risk.

All our suggested risk-adapted starting age of PCa screening are based on the age at which the risk of stage III/IV and/or lethal PC reaches the same level as such risk for men at benchmark starting age of screening in the general population. A sensitivity analysis was conducted to compare results for period 1990 to 2015 (after introducing PSA testing) and for the whole study period (1958 to 2015). All analyses were performed using SAS 9.4 (SAS Institute, Cary, North Carolina, US). The study protocol was approved by the Lund Regional Ethics Committee (2012/795). Pseudonymized data were used for our analyses. This study is reported as per the REporting of studies Conducted using Observational Routinely-collected health Data (S1 RECORD Checklist) guidelines [31]. The analyses were planned ahead, and no data-driven changes to analyses took place (S1 Analysis Plan). Some new analyses were conducted as requested by reviewers (e.g., 95% CIs, comparison of results for period 1990 to 2015 with the ones for the whole study period, and risk of PCa for men with one affected FDR who had 0/1/more brothers without PCa).

## Results

During the follow-up of 6,343,727 men (aged 0 to 96 years at baseline), a total of 88,999 patients were diagnosed with stage III/IV PCa or died due to PCa. Taking the dynamic nature of family history into account, 4.3% of patients with stage III/IV or fatal PCa had at least 1 FDR with PCa before the time of their diagnosis, whereas using the static approach, 13% of patients with stage III/IV or fatal PCa had at least 1 FDR with PCa before the time of their diagnosis. The risk of being diagnosed with stage III/IV or fatal PCa in the next 10 years for men at age 45 in the general population was 0.1%; at age 50, this risk was 0.2%, at age 55, 0.6%, and at age 60, 1.3%.

## Risk by number of affected relatives

The 10-year cumulative risk of invasive PCa increased with increasing age and number of affected relatives (Figs 1 and 2). For example, at age 50 (a common benchmark starting age of screening for PCa), the 10-year cumulative risk of being diagnosed with stage III/IV PCa or fatal PCa for men in the general population was 0.2% (Table 1). Men with only 1 affected FDR reached this screening level of risk at age 46, and those with ≥2 affected FDRs, at age 41. For people with no family history of PCa, their risk of stage III/IV or fatal PCa was slightly lower than the general population (Fig 1). Similar patterns were seen in other benchmark starting ages of PCa screening (Table 1). An additional analysis was done to compare the risk of PCa for men with 1 affected FDR who had 0/1/more brothers without PCa, and the results showed that the number of unaffected brothers does not change the results substantially (S1 Fig).

## Risk by age at diagnosis in relatives

Younger age at diagnosis of PCa in FDRs was associated with higher risk of stage III/IV or fatal PCa in other family members (Figs 1 and 2). For example, for screening benchmark starting age of 50 years, the age of reaching screening level risk for those with only 1 affected FDR diagnosed before age 60 was 7 years earlier than the general population; for those with 1 affected FDR at age 60 to 69, 5 years earlier; with 1 FDR diagnosed at age 70 or older, 3 years earlier; with ≥2 FDRs before age 60, 9 years earlier; and with ≥2 FDRs at age 60 or older, 7 years earlier (Table 1). Similar patterns were seen in other benchmark starting ages of screening.

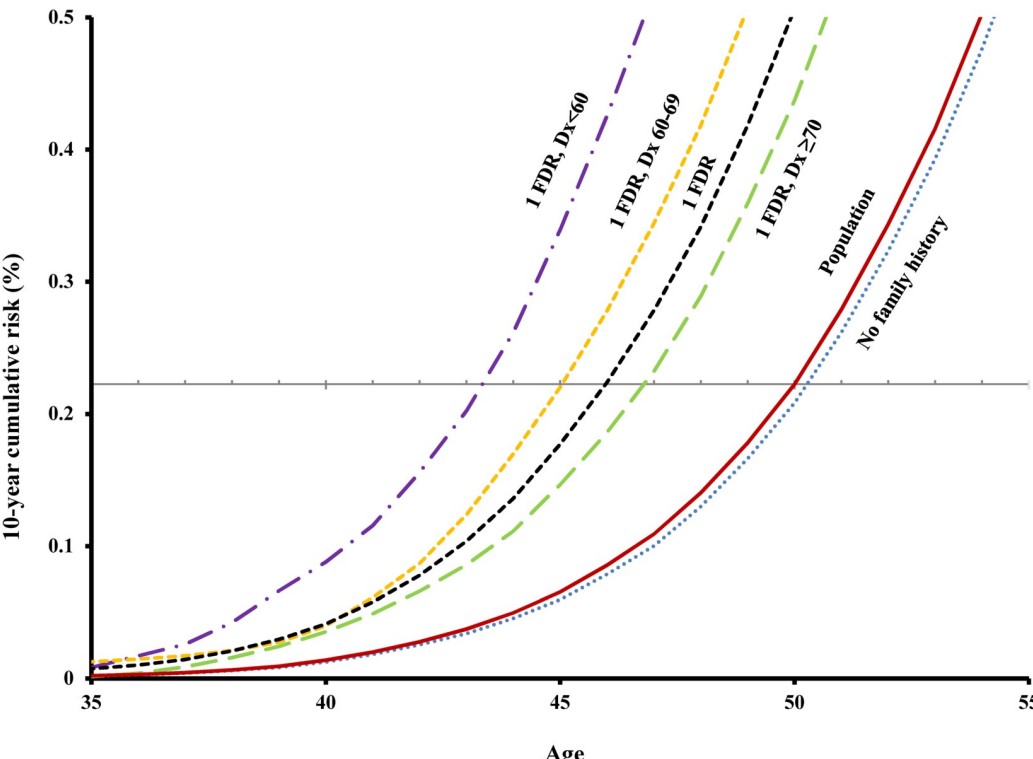

**Fig 1. Age-specific 10-year cumulative risk of stage III/IV PCa or fatal PCa by age at diagnosis of invasive PCa in the affected FDR.** The gray horizontal line corresponds to 10-year cumulative risk level for 50-year-old men in the population. Dx, diagnosis; FDR, first-degree relative; PCa, prostate cancer.

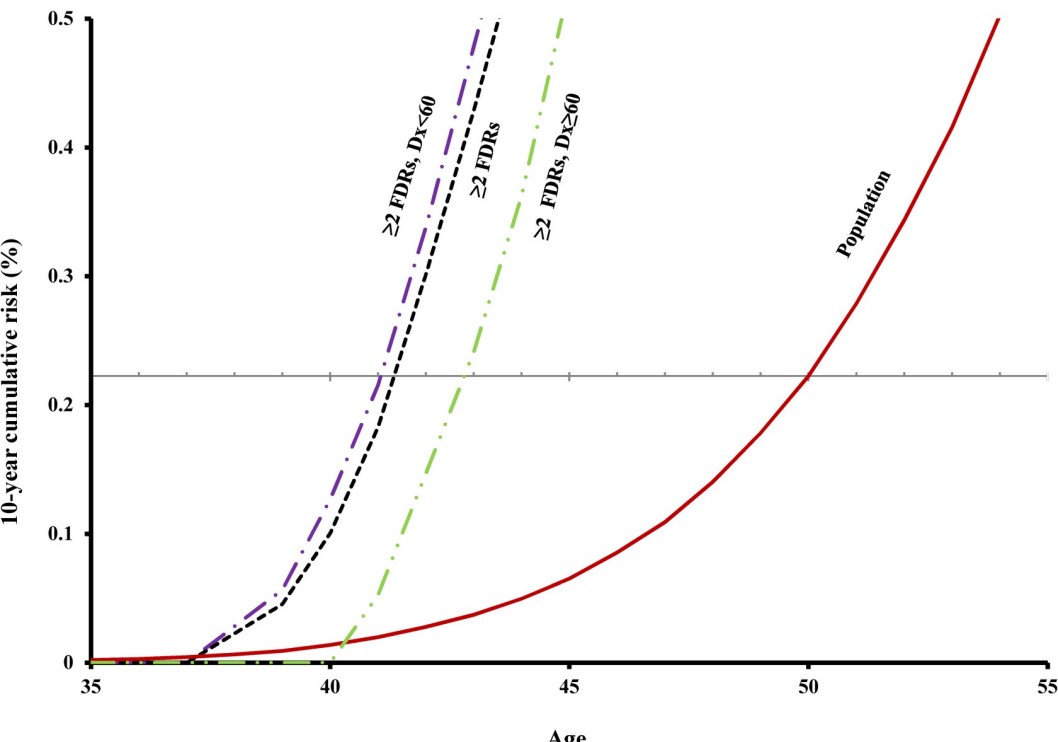

**Fig 2. Age-specific 10-year cumulative risk of stage III/IV or fatal PCa by youngest age at diagnosis of invasive PCa in men with ≥2 affected FDRs.** The gray horizontal line corresponds to 10-year cumulative risk level for 50-year-old men in the population. Dx, diagnosis; FDR, first-degree relative; PCa, prostate cancer.

## Risk by type of affected relatives (father versus brother)

We additionally conducted the analyses stratified by type of affected relatives (father versus brother; S1 Table), and the results were quite similar. For instance, when recommended benchmark starting age of PCa screening in the general population was 50 years,

**Table 1. Risk-adapted starting age of PCa screening for different benchmark starting ages of screening by number of affected FDRs and youngest age at diagnosis of relatives using 10-year cumulative risk.**

| Family history | Age at diagnosis of youngest relative, y | Cancer patients (*N*) | Risk-adapted starting age of screening, y (95% CI) | | | |
|---|---|---|---|---|---|---|
| | | | **[45]** | **[50]** | **[55]** | **[60]** |
| **Population [benchmark age]** | - | 88,999 | [45] | [50] | [55] | [60] |
| **1 FDR** | All ages | 3,576 | 41 (41–42) | 46 (45–46) | 51 (50–51) | 56 (55–56) |
| | <60 | 449 | 39 (37–41) | 43*(41–45) | 48 (46–50) | 53 (52–55) |
| | 60–69 | 1,117 | 41 (40–42) | 45 (44–46) | 50 (49–51) | 54 (54–55) |
| | ≥70 | 2,010 | 42 (41–43) | 47 (46–47) | 52 (51–52) | 56 (56–57) |
| **≥2 FDRs** | All ages | 311 | 39 (39–42) | 41 (39–45) | 44 (42–47) | 48 (46–51) |
| | <60 | 120 | 39 (38–42) | 41 (39–44) | 44 (41–47) | 48 (44–51) |
| | ≥60 | 191 | 41 (40–45) | 43 (41–47) | 46 (43–49) | 49 (45–52) |
| **10-year cumulative risk in the general population** | | | 0.1% | 0.2% | 0.6% | 1.3% |

Bold ages 45, 50, 55, and 60 indicate benchmark starting ages of PCa screening in the general population.

*Example: When recommended benchmark starting age of PCa screening in the general population was 50 years, men who had a history of PCa diagnosed before age 60 in 1 FDR attained the same risk level of 50-year-old men in the general population (0.2%) at age 43 and thus they could start screening 7 years earlier.

CI, confidence interval; FDR, first-degree relative; PCa, prostate cancer.

men who had a history of PCa diagnosed between age 60 and 69 only in 1 brother or only in his father attained the same risk level of 50-year-old men in the general population at age 45.

## Fifteen-year and 20-year cumulative risks

We additionally conducted the analyses using 15-year and 20-year cumulative risks, which resulted in quite similar ages of reaching the same level of risk of stage III/IV and/or lethal PCa as risk in 50-year-old men in the general population (S2 and S3 Tables). For instance, when recommended benchmark starting age of PCa screening in the general population was 50 years, men who had a history of PCa in 1 FDR diagnosed between age 60 and 69 attained the same corresponding 10-year, 15-year, and 20-year cumulative risk levels of 50-year-old men in the general population at age 45.

## Comparison with recommendations by guidelines

We also compared our evidence-based risk-adapted starting ages of screening for PCa with mostly experts' opinion-based starting ages of PCa screening recommended by current guidelines (Table 2). Our risk-adapted starting ages of PCa screening, which were based on the consideration of history of PCa in FDRs and their age at diagnosis, were substantially (up to 11 years) younger than the general starting ages for PCa screening recommended by different guidelines. For example, when recommended benchmark starting age of PCa screening in the general population was 55 years, men with 2 FDRs diagnosed with PCa after age 60 attained the same risk level of 55-year-old men in the general population at age 46. This was 9 years earlier than the recommended age by American Urological Association (AUA) Guideline and the PCa screening recommendation by the USPSTF.

**Table 2. Comparison between our evidence-based risk-adapted starting age of screening and recommended age of PCa screening by different guidelines.**

| Family history | Youngest relative's age at diagnosis (years) | Recommended starting age (years) in population→ 45 | | | 50 | | | 50 | | | 55 | | |
|---|---|---|---|---|---|---|---|---|---|---|---|---|---|
| | | NCCN | Evidence[1] | Diff.[2] | EAU, CUA, and ACP | Evidence[1] | Diff.[2] | ACS | Evidence[1] | Diff.[2] | AUA and USPSTF | Evidence[1] | Diff.[2] |
| **1 FDR** | **<60** | 45 | 39 | 6 | 45 | 43 | 2 | 45 | 43 | 2 | 55 | 48 | 7 |
| | **60–69** | 45 | 41 | 4 | 45 | 45 | 0 | 50 | 45 | 5 | 55 | 50 | 5 |
| | **≥70** | 45 | 42 | 3 | 45 | 47 | −2 | 50 | 47 | 3 | 55 | 52 | 3 |
| **≥2 FDRs** | **<60** | 45 | 39 | 6 | 45 | 41 | 4 | 45 | 41 | 4 | 55 | 44 | 11 |
| | **≥60** | 45 | 41 | 4 | 45 | 43* | 2 | 50 | 43* | 7 | 55 | 46 | 9 |

[1] The recommended evidence-based risk-adapted starting age of screening from our study.

[2] Difference: age recommended by guideline minus our evidence-based risk-adapted age.

*Example: When recommended benchmark starting age of PCa screening in the general population was 50 years, men with 2 FDRs diagnosed with PCa after age 60 attained the same risk level of 50-year-old men in the general population at age 43 and thus they could start screening 7 years earlier. This was 7 years earlier than the recommended age by ACS and 2 years earlier than that of EAU, CUA, and ACP.

ACS, American Cancer Society guideline for the early detection of prostate cancer: update 2010; ACP, Screening for prostate cancer: a guidance statement from the Clinical Guidelines Committee of the American College of Physicians; AUA, American Urological Association Guideline; CUA, Canadian Urological Association recommendations on prostate cancer screening and early diagnosis; EAU, European Association of Urology (EAU)—European Society for Radiotherapy & Oncology (ESTRO)—International Society of Geriatric Oncology (SIOG) EAU-ESTRO-SIOG Guidelines on Prostate Cancer; FDR, first-degree relative; NCCN, The National Comprehensive Cancer Network Clinical Practice Guidelines in Oncology for Prostate Cancer Early Detection Version 2.2016; PCa, prostate cancer; USPSTF, Screening for Prostate Cancer: US Preventive Services Task Force Recommendation Statement.

## Sensitivity analysis

A sensitivity analysis by calendar period at diagnosis comparing the results for the period 1990 to 2015 to the ones for the whole period (1958 to 2015) was conducted, and the analysis showed consistent results (S2 Fig).

# Discussion

## Findings of the study

Our study estimated the risk of stage III/IV or fatal PCa in family members of patients with PCa. Furthermore, we supplied practical information on the risk-adapted starting age of screening for family members of patients with PCa. We showed that age, age at diagnosis of PCa in relative/s, and number of affected FDRs are important elements in increased risk of stage III/IV or fatal PCa, and these factors accordingly resulted in differing risk-adapted starting ages of PCa screening. We calculated how many years earlier men with a history of PCa in their FDRs reach the same screening level of stage III/IV or fatal PCa risk as their peers in the general population. We also provided risk-adapted starting ages for other benchmark screening ages, such as 45, 55, and 60 years, to fit in different population with different risk level of stage III/IV or fatal PCa. This large-scale cohort study showed that relatives of patients with PCa reached the screening level of risk up to 11 years earlier than the recommended ages of PCa screening by current guidelines in North America and Europe [9,12–18].

## Comparisons with other studies

We found that early-onset PCa in relatives and higher number of affected close relatives are associated with increased risk of stage III/IV or fatal PCa, which are in line with previous studies on familial risk of invasive PCa without considering the stage at diagnosis [32–35]. However, by assessing the familial risk only for stage III/IV or fatal PCa, we tried to mitigate the increased familial risk due to overdiagnosis because diagnosis of PCa in a man with an aggressive outcome after diagnosis of PCa in a relative cannot be attributed to overdiagnosis. Furthermore, previous studies provided no practical information for personalized age of initiation of PCa screening based on different constellations of family history. We calculated the risk-adapted starting age of PCa screening based on our familial risk estimates driven from the world's largest nationwide family-cancer datasets. It has been reported before that the number of relatives (family size) does not affect PCa risk [36]. However, an additional analysis was conducted to compare the risk of PCa for men with 1 affected FDR who had 0/1/more brothers without PCa diagnosis, and the results showed similar estimated risk-adapted starting ages of screening by number of unaffected brothers.

The risk of developing familial cancer has been proposed to depend both on a person's own age and on the age at which their relatives receive a diagnosis of a concordant cancer [34]. Familial cancers are early onset mainly in those individuals whose family members are affected at early ages [34]. This dual dependence on age was also evident in our results, as the highest familial risk of early-onset stage III/IV or fatal PCa was observed in patients whose FDRs received a PCa diagnosis at younger ages. It is possible that the age at diagnosis of cancer in family members could be affected by PSA screening, for which we had no data. No national population-based PSA screening program has ever been recommended in Sweden, and organized PSA testing was implemented in some parts of Sweden only since 2019 [37]. Despite this, an additional sensitivity analysis was performed to compare risk estimations over different calendar periods, and the results were consistent. A recent study has shown that even family history of prostate borderline or in situ carcinoma (mostly screen-detected lesions) and

their age at diagnosis are rather equally important and should be taken into account in counseling patients and their relatives [38].

To our knowledge, the American Cancer Society (ACS) Recommendations for Prostate Cancer Early Detection is the only official guideline that currently takes both the number of affected relatives with PCa and age at diagnosis of PCa in the relative into account. It recommends that men who have an FDR diagnosed with PCa before age 65 should start screening 5 years earlier than average-risk men in the general population [15]. In general, our results based on nationwide data showed that risk-adapted starting age of screening for relatives of patients with PCa differs with different family constellation and age at diagnosis of PCa in their relatives and could be up to 11 years earlier than current recommendations. Although the incidence rate of PCa varies by geography and race/ethnicity, the familial risks of cancers are generalizable across populations with approximately similar cancer incidence and pattern [39]. Thus, the risk advancement time (the difference between the age at which the whole male population and the age at which men with a particular family history constellation attain screening-level risk) is likely to be similarly generalizable. Nevertheless, it would be optimal for the risk-adapted starting ages of screening to be externally validated in countries with population compositions different from Sweden. Besides, we also provided risk-adapted starting ages of screening for populations at different benchmark starting ages of screening, which could be used in countries with other benchmark ages than 50 years.

Another commonly neglected issue in previous familial PCa studies is the time-dependent nature of family history in the real world. Generally, family history of PCa has been dealt with as a static variable (having ever a family history of PCa or not) in spite of the dynamic nature of family history, which changes over time when a new family member is affected. In our study, we used the dynamic definition of PCa family history, which means that changes in status of PCa family history during the lifetime of an individual have been taken into account, making our results more meaningful for defining risk-adapted starting ages of screening in a real-world setting. One should note that using the dynamic method of family history assessment has nothing to do with or against the constant predisposing genes in families; it is only a method to derive more accurate estimates in risk prediction studies to be used for risk stratification in clinic.

Individuals with a family history of PCa, irrespective of number of affected relatives with PCa and age at diagnosis of PCa in the relative, have been recommended to be screened 5 years earlier than the proposed year of screening for the general population by 3 guidelines [12–14]. This recommendation was based on one study from a population-based registry in Utah state in the US [12–14,33]. However, in that study, which included 636,443 men (10% of our study population), only relative risk estimates were calculated under the static definition of family history, and no starting age of screening was provided, similar to other previous studies on familial risk of PCa that were not designed to provide risk-adapted starting age of screening [32–35]. In a real-world scenario, family history is not fixed, but changes during lifetime. However, this time-varying nature of family history has been commonly ignored in studies on PCa familial risk.

## Strengths and weaknesses

Apart from novel approaches in this study, which resulted in firsthand information that could be used for risk-adapted (personalized) starting age of PCa screening, we used the largest familial cancer datasets available to our knowledge, and our analyses considered the number of relatives affected, age at diagnosis of relatives, and timing of the cancer events in relatives in a more comprehensive way than in previous studies [32–35]. We used only risk of stage III/IV

or fatal PCa as the outcome to mitigate the overestimation of familial risk due to aggregation of indolent (screen-detected) cancers in families. Another important advantage of our study was the accuracy and completeness of the analyzed datasets, mitigating biases related to over- and underreporting of family history, selection, and recall biases, as we utilized register-based family history (not patients' self-reports) and medically verified cancer status from long-standing nationwide cancer registry data. Moreover, we also calculated the 15- and 20-year cumulative risks of advanced PCa with consistent estimates for starting ages, which adds to the robustness of our results.

We had no data on genetic information; however, family history, which represents familial risk stemming from shared genetic and shared environmental factors, is still the strongest risk factor for PCa that plays an important role in risk evaluation and decision-making about initiation of PCa screening [2,40,41]. The level of screening affects the proportion of men with a family history, which, in turn, would affect the proportion of men having to be screened at earlier ages. It could also affect the excess risk of stage III/IV or fatal cancer associated with a family history, since with screening a higher proportion of cases in relatives would be indolent. Thus, the excess risk of stage III/IV or fatal PCa associated with a family history could be lower in a highly screened population. Sweden with PCa mortality higher than many other developed countries is not considered as a highly screened country [1]. PSA testing prevalence has previously been shown to vary across different regions of Sweden [42,43]. Data from Stockholm region showed that, during 2010 and 2011, 25%, 40%, and 46% of men aged 50 to 59, 60 to 69, and 70 to 79 had a PSA test, respectively, whereas less than 20% of the male population underwent PSA testing in a large healthcare region in Sweden [43,44]. The stage III/IV or lethal familial risk estimations could have been diluted when men with family history of PCa were diagnosed early due to actively seeking earlier PSA screening and did not die of PCa compared to men without family history, although a previous study has reported that familial aggregation of PCa is not due to screening habits shared in a family [45]. Screening practice may change the reference group in calculation of familial relative risk of aggressive PCa (e.g., in terms of standardized incidence ratio). However, our results based on cumulative risk of aggressive PCa that has no reference group is not affected by screening practice, and this explains the stable results over different periods in the sensitivity analysis that we conducted. Another limitation of this study was the rather homogeneity of Swedish population, which did not allow stratified analyses by race/ethnicity. However, as different starting ages of screening are (or should be) recommended for different races/ethnicities or countries, the possible risk-adapted starting ages of screening for populations at different benchmark starting ages of screening were provided to fit in with races/ethnicities/countries that have different risk levels of PCa.

## Implications of findings

Family history as the strongest risk factor for PCa plays an important role in risk evaluation and decision-making about initiation of PCa screening [2,40,41]. However, the latest USPSTF in 2018 was not able to make a separate, specific recommendation on PSA-based screening for PCa in men with a family history of PCa, mainly due to lack of evidence-based information in this regard and the potential to increase more harm than benefit due to overdiagnosis of PCa [9]. Vickers and Lilja demonstrated that one can identify men at increased risk for advanced PCa at early middle age [46]. They suggested that most advanced cancers can be detected with sufficient lead time to allow curative therapy. A breakthrough paper suggested that risk-adapted PSA screening based on an early PSA value in men 45 to 49 year of age were correlated to metastasis or death from PCa about 25 to 30 year later [47]. A rather recent review suggested

that screening offers a benefit in reducing PCa mortality. The controversy surrounding the screening of PCa is that most screening-detected cancers are indolent or of early stage without much impact on PCa mortality, and by current PCa screening strategies, screening numbers needed to prevent one death due to PCa is too high [20,48,49]. Therefore, screening strategies for PCa should be shifted to target men at high risk of developing aggressive PCa to reduce mortality from this cancer. Any screening recommendations must consider the false positives and potential harms from screening individuals earlier. Side effects of longer treatment and extended active surveillance associated with earlier screening should be also taken into account. Screening programs must weigh both the sensitivity and specificity. Familial aggregation of both incident and fatal PCa had been observed, and in fact, it has been suggested that starting screening before any PCa is diagnosed in a family appears not to be warranted [41,45]. However, considering a large proportion of men for whom the family history-based risk-adapted screening does not apply, further investigation by incorporating additional information on common variants and other risk factors is important to improve PCa screening and prevention in a more personalized manner.

A rather elaborate anamnesis of family history is important in implementing the findings of this study into clinical practice. A simple question—"How many relatives if any had PCa in family" would be insufficient for this purpose. As shown by our data, higher number and younger age at diagnosis of affected FDRs are both important factors associated with increased risk of aggressive/fatal PCa. A pilot test of oncological practices showed complete family history report rates of 77% in FDRs and 61% in second-degree relatives, but age at diagnosis was recorded in only one-third of them [50]. In order to benefit from results of this study, detailed family history of close relatives (father, brother, and son) would be needed along with the age at diagnosis of PCa in them (<60, 60 to 69, ≥70 years), which is of course easier to obtain than the exact age at diagnosis for each affected family member.

The American Society of Clinical Oncology guideline also recommends that emphasis should be put on careful documentation of the family history of close relatives [51]. Self-reports can still serve as an important source of information in clinical counseling since only information of close relatives would be needed. For countries with mature registry (or electronic health record) system like Sweden, family history could be technically taken from register-based datasets, although, to our knowledge, this is not the case in any country. A recent practice improvement project at Mayo Clinic showed that enhanced family history screening was a low cost but effective method for individualized cancer surveillance [52]. Various tools for family history collection are being developed, and systematic reviews have shown a 46% to 78% improvement in data collection [53,54]. As an attempt to alleviate the burden of physicians, health policy makers could remove the system barriers and facilitate the development and implementation of family history collection tools (e.g., by providing websites, software, and/or apps to collect the necessary family history before the actual visit of men to the physician). Adequate health education to the lay population seems also very important for timely assessment of risk profile. Health economic studies are warranted to encourage insurances to cover cost of evidence-based personalized early screening in high-risk men.

As an effort toward individualized screening, in contrast to the past one-size-fits-all strategy, based on findings of this study, we suggest to divide men with family history into 5 diversified groups to be able to propose more accurate screening and prevention interventions according to their risk levels. The goal of precision medicine and the utilization of family history in the disease risk assessment are highly aligned [54]. Further researches are warranted about practicalities of implementing screening based on family history and comparison of cost-effectiveness of risk-adapted screening versus that of current age-based screening practice.

The risk-adapted starting age of PCa screening for men with a family history of PCa ranged from 3 to 12 years earlier than men in the general population according to the number of affected FDRs and youngest age at diagnosis of relatives. Clinician can inform patients with PCa about the possible earlier screening in their FDRs. Patients can encourage their close relatives to seek early counseling with their doctor to find out the appropriate age of initial PCa screening for them based on their specific detailed family history. Country-specific clinical guidelines need to recommend men to start PCa screening at different ages based on their family history and age at diagnosis of PCa in their families. Results of our study (summarized in Table 1) can be easily used by clinicians based on answers to the following 2 simple questions:

1) How many of your close relatives (father, brothers, or sons) have ever received a prostate cancer diagnosis?

2a) If the answer to first question is 1, then ask:

   Was the relative diagnosed before age 60 years, between 60 and 69, or at 70 or older?

2b) If the answer to first question is 2 or more, then ask:

   Was any of relatives diagnosed before age 60 years?

Of course, if the family history is going to be used for future research, it would be ideal to record exact number of affected relatives, their ages at diagnoses, and calendar years of their diagnoses when possible [26]. As an example, according to Table 1, when recommended benchmark starting age of PCa screening in the general population in a country is 50 years, men who had 1 FDR diagnosed with PCa before age 60 could start screening at age 43, i.e., 7 years earlier than his counterparts in the general population. If there were 2 FDRs with PCa in the family of that man, one diagnosed before age 60 and the other diagnosed after age 60, the youngest age at diagnosis would be before age 60, and therefore, the man could be screened at age 39, i.e., 11 years earlier.

## Conclusions

In conclusion, this study provides novel information that could be used for risk-adapted starting age of screening in FDRs of patients with PCa based on the world's largest nationwide register-based family-cancer datasets. The results can contribute to evidence-based personalized PCa screening guidance. Clinicians could inform patients with PCa about this possibility and encourage personalized counseling for their relatives. Our findings provide practical information to supplement current guidelines for PCa screening.

## Supporting information

**S1 RECORD Checklist. The RECORD statement—checklist of items, extended from the STROBE statement, which should be reported in observational studies using routinely collected health data.** RECORD, REporting of studies Conducted using Observational Routinely-collected health Data; STROBE, Strengthening The Reporting of OBservational Studies in Epidemiology.
(DOCX)

**S1 Fig. Age-specific 10-year cumulative risk of stage III/IV PCa or fatal PCa by family size (number of brothers in family) in men with 1 affected FDR.** The gray horizontal line corresponds to 10-year cumulative risk level for 50-year-old men in the population. FDR, first-degree relative; PCa, prostate cancer.
(TIF)

**S2 Fig. Age-specific 10-year cumulative risk of stage III/IV PCa or fatal PCa in men with 1 or ≥2 affected FDRs by calendar period (1958–2015 and 1990–2015).** The gray horizontal line corresponds to 10-year cumulative risk level for 50-year-old men in the population. FDR, first-degree relative; PCa, prostate cancer.
(TIF)

**S1 Table. Risk-adapted starting age of PCa screening for different benchmark starting ages of screening by type of affected relatives and age at diagnosis of the relative using 10-year cumulative risk.** PCa, prostate cancer.
(DOCX)

**S2 Table. Risk-adapted starting age of PCa screening for different benchmark starting ages of screening by number of affected relatives and youngest age at diagnosis of FDRs using 15-year cumulative risk.** FDR, first-degree relative; PCa, prostate cancer.
(DOCX)

**S3 Table. Risk-adapted starting age of PCa screening for different benchmark starting ages of screening by number of affected relatives and youngest age at diagnosis of FDRs using 20-year cumulative risk.** FDR, first-degree relative; PCa, prostate cancer.
(DOCX)

**S1 Analysis Plan. Statistical analysis plan.**
(DOCX)

## Author Contributions

**Conceptualization:** Elham Kharazmi, Mahdi Fallah.

**Data curation:** Mahdi Fallah.

**Formal analysis:** Xing Xu, Mahdi Fallah.

**Funding acquisition:** Mahdi Fallah.

**Investigation:** Elham Kharazmi.

**Methodology:** Elham Kharazmi, Yu Tian, Trasias Mukama, Mahdi Fallah.

**Resources:** Kristina Sundquist, Jan Sundquist, Mahdi Fallah.

**Supervision:** Elham Kharazmi, Mahdi Fallah.

**Writing – original draft:** Xing Xu, Elham Kharazmi, Mahdi Fallah.

**Writing – review & editing:** Xing Xu, Elham Kharazmi, Yu Tian, Trasias Mukama, Kristina Sundquist, Jan Sundquist, Hermann Brenner, Mahdi Fallah.

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
