## [Editor Report · Decision Letter 0]

23 Jun 2020

Dear Dr Fallah, 

Thank you for submitting your manuscript entitled "Risk-adapted starting age of screening in relatives of patients with prostate cancer: Real world evidence from a nationwide cohort to supplement current screening recommendations" for consideration by PLOS Medicine.

Your manuscript has now been evaluated by the PLOS Medicine editorial staff and I am writing to let you know that we would like to send your submission out for external peer review.

Kind regards,

Helen Howard, for Clare Stone PhD 

Acting Editor-in-Chief

PLOS Medicine 

plosmedicine.org

---

## [Decision Letter · Decision Letter 1]

5 Aug 2020

Dear Dr. Fallah,

Thank you very much for submitting your manuscript "Risk-adapted starting age of screening in relatives of patients with prostate cancer: Real world evidence from a nationwide cohort to supplement current screening recommendations" (PMEDICINE-D-20-02475R1) for consideration at PLOS Medicine. 

Your paper was evaluated by a senior editor and discussed among all the editors here. It was also evaluated by five independent reviewers, including a statistical reviewer. The reviews are appended at the bottom of this email and any accompanying reviewer attachments can be seen via the link below:

[LINK]

In light of these reviews, I am afraid that we will not be able to accept the manuscript for publication in the journal in its current form, but we would like to consider a revised version that addresses the reviewers' and editors' comments. Obviously we cannot make any decision about publication until we have seen the revised manuscript and your response, and we plan to seek re-review by one or more of the reviewers. 

We expect to receive your revised manuscript by Aug 26 2020 11:59PM. Please email us (plosmedicine@plos.org) if you have any questions or concerns.

We look forward to receiving your revised manuscript. 

Sincerely,

Emma Veitch, PhD

PLOS Medicine

On behalf of Clare Stone, PhD, Acting Chief Editor,

PLOS Medicine

plosmedicine.org

*Within the Abstract Methods and Findings section, please include a brief note about any key limitation(s) of the study's methodology.

*At this stage, we ask that you include a short, non-technical Author Summary of your research to make findings accessible to a wide audience that includes both scientists and non-scientists. The Author Summary should immediately follow the Abstract in your revised manuscript. This text is subject to editorial change and should be distinct from the scientific abstract. Please see our author guidelines for more information: https://journals.plos.org/plosmedicine/s/revising-your-manuscript#loc-author-summary

*Please clarify if the analytical approach followed in this study corresponded to one laid out in a prospective protocol or analysis plan. Please state this (either way) early in the Methods section.

*We'd recommend the authors use an appropriate reporting guideline to support reporting of their study - one such guideline may be the RECORD guideline (https://www.equator-network.org/reporting-guidelines/record/) - designed for routinely collected health data. If using this guideline please cite the RECORD paper in the Methods section and also upload a completed RECORD checklist as supporting information with your revision. 

Comments from the reviewers:

Reviewer #1: I confine my remarks to statistical aspects of this paper. These were well done and I have only a couple of minor corrections/suggestions.

p 3 Give the death rate for prostate cancer. "5th leading cause ...." is pretty meaningless. 

p. 5 First para. Men don't necessarily learn about a relative's PC diagnosis.

p. 5 Please format the formulas better, maybe separating them from the text

Peter Flom

Reviewer #2: Manuscript#: PMEDICINE-D-20-02475R1

Title: Risk-adapted starting age of screening in relatives of patients with prostate cancer: Real world evidence from a nationwide cohort to supplement current screening recommendations. 

Criteria PLOS Medicine (website consulted on 04.07.2020):

General:

1. The research question is an important one to the community of researchers in this general area.

2. The results provide a substantial advance over existing knowledge, with clear implications for patient care, public policy, or clinical research agendas.

3. Published together with an Author Summary written for general readers, the article is of interest to clinicians and policymakers who are not specialists in this topic.

* What are the main claims of the paper and how significant are they for the discipline?

* Are the claims properly placed in the context of the previous literature? Have the authors treated the literature fairly?

* Do the data and analyses fully support the claims? If not, what other evidence is required?

* PLOS Medicine encourages authors to publish detailed methods as supporting information online. Do any particular methods used in the manuscript warrant such publication? If a protocol is already provided, for example for a randomized controlled trial, are there any important deviations from it? If so, have the authors explained adequately why the deviations occurred?

* Is this paper outstanding in its discipline? If yes, what makes it outstanding? If not, why not?

* Does the study conform to any relevant guidelines such as CONSORT, MIAME, QUORUM, STROBE, and the Fort Lauderdale agreement?

* Are details of the methodology sufficient to allow the experiments to be reproduced?

* Is any software created by the authors freely available?

* Is the manuscript well organized and written clearly enough to be accessible to non-specialists?

Summary:

In the revised manuscript the authors analyze the complete Swedish men population in a longitudinal observation between 1958 and 2015 with the goal to study the relatives having had advanced PCa or who succumbed to PCa for purpose of calculating the risk-adapted starting age of screening for PCa for their relatives. They calculate the 10-year cumulated PCa risk in the whole population (0.2%) as a reference cutoff value and test it against various populations build on criteria of number of relatives with PCa and age of PCa diagnosis. They demonstrate that these populations reach this risk cutoff at the different age and thus they propose tailoring the onset of PCa screening accordingly. 

This topic is very relevant for everyday practice of every clinician dealing with early detection of PCa, thus all three general criteria of PLOS Medicine as listed above are fulfilled. 

As the authors properly describe in the introduction - a substantial body of literature exists on the topic and there is a clear agreement that positive family history increases risk of having PCa in significant manner, thus these men should receive more screening for PCa and at earlier age. However, the proposed cutoffs vary and if you consult medical guidelines (e.g. EAU or ESMO) these cutoffs can be perceived as an extrapolated and rather intuitive best guess concerning the starting age of screening. With this important study the authors try to fill this gap in more precise manner. Moreover they take into account not only the number of relatives but also onset of disease in those relatives, which is additional important piece of information. 

Comments: 

Introduction

Is well written. No relevant remarks. 

Methods 

Material for the study is well chosen and we can in general commend our Swedish colleagues for the quality registry data which have been collected. The statistical methods seem adequate and sufficient, at least at the understanding of reviewer who is clinician but with experience dealing with epidemiological and statistical methods and data. Still as the reviewer is not specialist in statistics a statistical review should be warranted. I would suggest to make a reference(s) to method source (can be a paper or manual) for the reader for better understanding. 

While the choice of the study event (advanced PCa or PCa death) seems correct as for purpose of the study some unclarities remain. Men who die at PCa are usually having PCa diagnosis before this event. Do I understand correctly that for men diagnosed with advanced PCa but dying from other causes this event was taken for study? It should be clarified as PCa diagnosis may be at advanced or local stage and then progress to PCa death (or not). Wouldn't it be more appropriate and consistent to take stage shift (or diagnosis) at advanced PCa stage as study event? 

For purpose of estimating the analysis reliability a reference (if available) to the reliability and completeness of both ascertainment of PCa death and disease stage in the registry should be done. 

Results

It is important that authors also calculated the 15- and 20-year cumulative risks for having PCa as this adds to the robustness of the results. 

Discussion 

Is generally well written but feels somehow uncritical. The authors see all but study strengths and omit the limitations. I have several remarks.

"Nevertheless, it would be optimal when risk-adapted starting ages of screening in other countries with different races than Sweden for men with a family history of prostate cancer are externally validated although the current experts' opinion-based recommendations for screening relatives of patients with prostate cancer are already in practice without validation."

This sentence is difficult to follow. As I understand the authors perceive their data as kind of validation or reference dataset. But from that perspective even if this may go with our intuition still their data have not been externally validated in another population. It is particularly important as PCa mortality in Sweden is particularly high as compared to the whole western world. Thus what happens in Sweden does not necessarily have to be happening elsewhere. The reference to "different races than Sweden" is also unclear. The reviewer presumes (but may be wrong) that the caucasian men is the predominant race in Sweden and that results apply to this race only. This paragraph should be elaborated. 

Study limitations are not mentioned. As already stated in previous comment, Swedish population has relatively high PCa specific mortality. While the methods used by the authors and the dynamic definition of positive family history may be generalized for other populations, the results of this study as such and the risk cutoffs may vary and might not be transferable to other countries or races. 

Another obvious limitation derives inherently from the fact that in the study only 4.3% of all men with PCa fulfilled their dynamic definition of family history and formed the study population. Thus the most men men doesn't have any family history. 

PCa diagnosis changed in time, while DRE and autopsy were method of choice at the study beginning, other modalities like PSA (and it's derivates or other biomarkers) , ultrasound, mpMRI, etc. This might have influenced the events distribution 

It is also important to convey to the reader that you can transfer these results only if you perform elaborate anamnesis of family history. A simple - "how many relatives if any had PCa in family" would be insufficient for that purpose. Has it been investigated how practicable and reliable is this approach in everyday GP practice? Checking register-based family history vs patients' self-reports would be interesting.

Minor remarks

"In general, our evidence-based results"

these results may become evidence if published thus "In general, our results" or "Our results" - whatever preferable. 

"Although the incidence rate of prostate cancer varies by geography and race/ethnicity, the familial risks of cancers are generalizable across populations with approximately similar cancer incidence and pattern."

Any reference to this general statement? 

Conclusion

The conclusion generalizes too far on "relatives of patients with PCa" as the authors explicitly chose advanced or lethal PCa as their study event. This self-limits the conclusions. 

Reviewer #3: The aim of the study is well defined. It is an interesting study dealing with the modalities of PCa screening for FDR in high risk families. It is known that family history is a significant risk factor for FDR increasing with the number of affected relatives in the family and early onset of the disease, this latter being a characteristic of inherited cancers. However, guidelines of screening in such families lack of precise recommendations according to the number of affected relatives and the onset of the PCa in the family. Moreover, as well underlined by the authors, screening in such families, and too early, may be associated with an increased risk of diagnosing indolent PCa and therefore overdiagnosis and overtreatment. It is the reason why the authors considered specifically the risk of late-stage/fatal risk of PCa which is a strength of the study. Using a large register-based nationwide cohort study, considered as the world's largest nationwide register-based family-cancer datasets, authors were able to provide very relevant results i.e. the 10-year (and also 15-year and 20-year) cumulative risk of invasive PCa increased according the number of affected relatives and by the youngest age at diagnosis in the family. These results can be considered as evidence-based risk-adapted starting ages of screening for PCa.

The conclusions are in agreement with the results.

The title is appropriate and the abstract provides significant and concise information. References are relevant and cited in the manuscript.

This is a very relevant well written paper and only few modifications should be done:

The authors should propose at the end of the discussion clear practical guidelines, as they would like to be supported by the different urologic and oncologic societies, on how to screen FDR in high risk families according number of affected relatives and the youngest affected in the family

Reviewer #4: This paper uses data from Swedish national registries to compute risk of aggressive prostate cancer (PCa) based on family history of PCa. Examining general population risk and risk for various family history profiles, the paper determines ages of starting screening that have equivalent risk of aggressive prostate cancer for those with a given family history as compared to the general population. More context is needed about the level of PSA screening in Sweden during the analysis period, since this could effect the various estimates. Also, more discussion about the practicalities of family history risk-based screening are needed.

Specific Comments

1. In Methods, the benchmark ages are given as 40,45,50 and 55. However, in Table 1, the benchmark ages are 45,50, 55 and 60.

2. Are there data on number of brothers? If so, is there any difference in risk for, say, a man with 1 first degree relative with PCa, if they had no brothers versus 1+ brothers (or 2+, etc.). More generally, could the exposure be refined based on number of male first degree relatives, in addition to how many had PCa. 

3. A potential harm of earlier start of screening is being diagnosed earlier with an indolent cancer and suffering from side-effects of treatment longer or having to undergo active surveillance longer. This should be mentioned in the discussion, in addition to harms from extra false positives that are mentioned in Discussion.

4. In order to interpret these data, it would be helpful to know the level of PSA screening in Sweden during the period of assessment. Presumably, it was low overall during the period linked to the cancer registry (1958-2015). The level of screening would certainly affect the proportion of men with a family history, which in turn would affect the proportion of men having to be screened at earlier ages. It could also affect the excess risk of stage III/IV or fatal cancer associated with a family history, since with screening a higher proportion of cases in relatives would be indolent, and indolent presentation could be in part genetically modulated. Thus the excess risk of stage II/IV or fatal disease associated with a family history could be lower in a screened population. This should be discussed.

5. The practicalities of implementing screening based on family history need to be discussed more. What is the expense and patient and physician time spent on evaluating family history, especially if it needs to be updated periodically? In most countries, family history data would not be available from national registries. How could EHR systems handle these data and could simple algorithms be developed to determine family history-based risk levels? Is there the potential for confusion in the population of the appropriate starting age if guidelines are based on family history, rather than a simple standard age?

Reviewer #5: Dear Dr Misra

This study provides new, clinically important, knowledge aand will be of interest for many specialities, but I'm not convinced that a general medical journal like PLoS Medicine is the best choice - a urology or cancer journal may be better. 

Moreover, the authors should present their data on prostate cancer risk as they are, rather than as direct evidence for start ages for screening. 

Finally, I should mention that I recently reviewed this manuscript for another journal (can't find my notes, though, and can't remember which one) and recommended similar changes as I do know, which means that the authors have not much revised their manuscript since then.

Yours sincerely

Ola Bratt

This is a large register-based cohort study on the influence of various combinations of family history (FH) of prostate cancer (PC) on the cumulative incidence of advanced/lethal PC. The analyses are well executed and the study provides clinically important, new knowledge, but the presentation of the results in the manuscript and the conclusions may be improved.

Major concerns: 

1) The calculated cumulative risks of Stage III/IV PC are clinically relevant, not least when considering recommendations about the start age of PC screening for men with a FH of PC, but it is not justified to present the results as "risk-tailored starting ages for PCa screening". The only way to provide evidence-based screening start ages for men with a FH of PC is through screening trials. There is absolutely no ground for claiming that the 10- or 15-year cumulative incidence of Stage III/IV PC are directly associated with the optimal starting age. The authors may of course discuss the use of their results for individualising the start age of PC screening and conclude that their results are suitable for this, but the proposed starting ages should not directly be presented as results. The manuscript needs to be re-written accordingly. Further examples of inappropriate phrasing are in the title ("Risk-adapted starting age of screening…"), the Abstract's conclusion ("an important step toward evidence-based personalized PCa screening guidance"), at the end of page 3 ("we aimed to provide risk-adapted starting age of screening for relatives of patients with PCa"; this could be changed to something like "we aimed to provide XXX that may be used for risk-adapting the starting ages…"), page 15 ("first-hand evidence-based information on risk-adapted (personalized) starting age of PCa screening") and in the final conclusion ("this study provides novel information on risk-adapted starting age of screening"). 

2) The suggested starting ages should rather be presented as something like "The age at which the risk of stage III/IV and/or lethal PC is the same as for men in the general population aged 50 years (0,2%)." Moreover, these ages should be presented with 95% confidence intervals.

3) Title: In addition to the above comment, also "Real world evidence" and "cohort" are misleading. "Cohort" may lead readers to believe that a cohort of screened men were observed. "Register study" may be better.

4) The limitations of the study are not well discussed. 4a) One is that the age of diagnosis, that is an important variable in the analysis, is highly dependent on when an individual's cancer is detected (localised or metastatic) and whether the cancer was detected after PSA testing or not (PSA testing confers up to 15 years lead time). A man's risk of PC is thus differently affected by a father diagnosed with a small low-grade PC after PSA-testing at age 70 versus a father diagnosed with metastatic disease at the same age. PSA-testing was introduced in Sweden in 1990 and became common in the late 1990s. This means that the age at diagnosis of relatives diagnosed before 1990 affects the PC risk differently than does the age at diagnosis of relatives diagnosed after year 2000.

4b) Another limitation is that the family history usually isn't revealed at many of the suggested start ages, so for instance supplementary table 2 describes a hypothetic scenario when it comes to the rows representing "Age at diagnosis of youngest relative" for brothers affected at 60-69 and > 70 years (few men aged 40 years have a brother aged over 60 years).

4c) Further limitations include that many men with a FH of PC obtain PSA testing and are diagnosed with stage 2 disease that is subsequently cured, but from which the man would have died had it not been detected and treated in time.

Minor comments

5) Methods, page 4: The statement "The Swedish National Board of Health and Welfare does not recommend a countrywide population-based PSA screening program for men" is not supported by reference 1.

6) Methods, page 4: The outcome was not the risk of late-stage/fatal prostate cancer. First, because occurrence of PC was the outcome, not the risk of this. Second, stage III includes poorly differentiated, localised PC, which is early-stage, not late-stage PC).

7) Methods, page 4: Most readers are probably not familiar with the AJCC staging system, so I recommend that stage III/IV is described in terms of TNM, Gleason score and PSA.

8) Results, 3rd sentence: The "static approach" has not been described in Methods. I assume this approach is incorrect, so why present results for this approach?

9) Page 8, 1st sentence under "Risk by age at diagnosis in relatives" is wrong. It should either be "The younger the age at diagnosis of PCa in FDRs, the higher the risk of..." OR

"Younger age at diagnosis of PCa in FDRs was associated with higher risk of...".

10) Page 9, 1st sentence: "rather comparable" is vague. Maybe the authors can substitute for a different term?

11) Discussion, end of 1st paragraph: RE "current guidelines, which are mostly based on the experts' opinion rather than evidence" - I'd say that the guidelines are based on the same kind of evidence that the present study provides, namely knowledge of an earlier age at onset of PC in men with a FH for PC, but by writing "current guidelines, which are mostly based on the experts' opinion rather than evidence", the authors imply that their own recommendations are strongly evidence-based, which they are not.

12) It is questionable whether the by the present study's authors' suggested highly variable starting ages are clinically more useful than the previous ones. Is it really realistic to produce clinical guidelines in which men with a FH of PC are stratified into 30 different groups with different recommended start ages? 

13) Page 13: Reference 26 reports deaths within 25-30 years, not 30-40 years.

14) The guidelines referred to from ACP, ACS & AUA (refs 13, 15, 16) are old (2010 - 2013). There must be newer versions.

Reviewer: Ola Bratt, Professor of Urology, Gothenburg University, Sweden

[LINK]

---

## [Decision Letter · Decision Letter 2]

23 Nov 2020

Dear Dr. Fallah,

Thank you very much for submitting your manuscript "Starting age of screening in relatives of patients with prostate cancer: Evidence from nationwide register-based cohort study to supplement current screening recommendations" (PMEDICINE-D-20-02475R2) for consideration at PLOS Medicine. 

[LINK]

In light of these reviews, I am afraid that we will not be able to accept the manuscript for publication in the journal in its current form, but we would like to consider a revised version that addresses the reviewers' and editors' comments. Obviously we cannot make any decision about publication until we have seen the revised manuscript and your response, and we plan to seek re-review by one or more of the reviewers. 

We expect to receive your revised manuscript by Dec 14 2020 11:59PM. Please email us (plosmedicine@plos.org) if you have any questions or concerns.

We look forward to receiving your revised manuscript. 

Sincerely,

Adya Misra, PhD

Senior Editor 

PLOS Medicine

plosmedicine.org

Title: study descriptor needed, remove starting age of screening etc. I suggest “Risk of cancer in relatives of patients with Prostate Cancer in Sweden: A cohort study” or similar 

Abstract- aim reads more like a recommendation. Need to revise this. There are recurring mentions of “providing risk adapted age of screening” however, I wonder if this is an implication of your findings and the work highlights the adapted risk of prostate cancer instead? Please revise throughout, as needed.

Abstract- age range of participants

Please add 2-3 limitations of your study in the last sentence of the abstract

Abstract conclusions- I suggest revising “highly valid”

Data availability- please add contact details of the relevant person(s) at Lund University who may be able to organise access to data for third parties

The author summary section “what did the researchers do” does not provide a summary of the research methods but its implementation in the clinic. Please remove the second and third bullet point and provide a summary of the methods and results

Please use quare brackets for references throughout 

Please provide the analysis plan as supplementary information and add a call out to this document in the methods section. In addition, please provide a completed RECORD checklist as supplementary information, using paragraphs and sections instead of page numbers

Page 12 “of PCa screening by current guidelines, which are mostly based on the experts’ opinion rather than evidence” this statement comes off rather combative and I suggest removing or rephrasing. Please also add a citation to the current guidelines, noting whether they are national (Swedish) or international guidelines

I suggest replacing all iterations of “high quality data” in the main text to be replaced by nationwide data or similar. I suggest organizing the Discussion as follows: a short, clear summary of the article's findings; what the study adds to existing research and where and why the results may differ from previous research; strengths and limitations of the study; implications and next steps for research, clinical practice, and/or public policy; one-paragraph conclusion.

The discussion is rather long and perhaps needs subheadings for readability. 

Page 14 “current experts’ opinion-based

recommendations for screening in relatives of patients with PCa are already in practice in many countries around the world without validation”. I suggest adding a reference or two in support or removing this sentence

Comments from the reviewers:

Reviewer #4: NO further comments

Reviewer #5: The authors have extensively revised and much improved the manuscript in accordance with the reviewers' numerous comments and suggestion. I must complement them on their thorough consideration of all the comments.

I have only one small suggestion, which the authors can feel free to adjust to or to discard, about their resopnse to my comment 5 Methods, page 4: 

The statement "The Swedish National Board of Health and Welfare does not recommend a countrywide population-based PSA screening program for men" is probably better supported by the following reference than by the dissertation.

Screening för prostatacancer. Rekommendation och bedömningsunderlag. Socialstyrelsen 2018. Access at https://www.socialstyrelsen.se/globalassets/sharepoint-dokument/artikelkatalog/nationella-screeningprogram/2018-10-15.pdf

[LINK]

---

## [Editor Report · Decision Letter 3]

3 Jan 2021

Dear Dr. Fallah,

Thank you very much for re-submitting your manuscript "Risk of prostate cancer in relatives of patients with this cancer in Sweden: A nationwide cohort study" (PMEDICINE-D-20-02475R3) for consideration at PLOS Medicine.

I have discussed the paper with our academic editor and it was also seen again by two reviewers. I am pleased to tell you that, provided the remaining editorial and production issues are dealt with, we expect to be able to accept the paper for publication in the journal.

[LINK]

We hope to receive your revised manuscript within about one week. Please email us (plosmedicine@plos.org) if you have any questions or concerns.

Please let me know if you have any questions. Otherwise, we look forward to receiving the revised manuscript soon.   

Sincerely,

Richard Turner, PhD

rturner@plos.org

Requests from Editors:

Please amend the title to "Risk of prostate cancer in relatives of prostate cancer patients in Sweden ...". 

Please begin the abstract "Evidence-based guidance ...".

In the abstract, please quote 95% CI around observations such as "9 years earlier" (you may prefer to state the age with associated 95% CI, as in table 1). 

In the abstract and elsewhere, please write "9 years earlier" rather than "nine years earlier", and so on, except at the start of sentences. 

Please revisit the phrase "parents' ages were not limited ..." in the abstract, which may need a few words of explanation - do you mean that full information on participants' fathers' lifespans was available in the study?

Please add a new final sentence to the "methods and findings" subsection of your abstract, which should begin "Study limitations include ..." or similar, and should list 2-3 of the study's main limitations. 

In the "Conclusion" subsection of your abstract, please adapt "can be used ..." to "could be used ...", to avoid overstatement of your findings' relevance in other populations. Please make a similar change in the final paragraph of the main text. 

Please delete the "keywords" section from the abstract page. 

In the author summary, please spell out "FDR" and "PCa" at first use. 

Please trim the central subsection of your author summary, which should consist of no more than 3-4 points of 1-2 short sentences each.

Please avoid "parent/s" in the author summary and any other instances in the paper. 

In the author summary, please amend the text to "... reach this screening risk threshold up to 11 years earlier ...". 

We notice that the author summary includes statements of "up to 11 years earlier" and "up to 12 years earlier". Please correct any inconsistency here or elsewhere in the paper. 

Please revisit the statement "the world's largest data" in your author summary. We suggest amending this to "Our study made use of the largest dataset available, to our knowledge, to identify the optimal age ..." or similar.

On p.14 of the paper, please make that "... we used the largest familial cancer datasets available, to our knowledge, and in our analyses considered ...".

Please revisit the "Strengths and weaknesses" subsection in the discussion. We suggest sumarizing the study's possible weaknesses in a discrete paragraph in a neutral way. At present, interspersed with statements such as "another important advantage", the discussion seems to avoid acknowledging possible limitations and implications for the study's conclusions. For example, are there possible differences in groups of different ethnicity?

Throughout the text, please remove spaces from the reference call-outs (i.e., "... of Sweden [47,48].").

Please delete the "author contributions" section from the end of the main text.

Please spell out the institutional author name for reference 9. 

Noting references 26, 28 & 45, for example, please ensure that all citations contain full access details.

We did not find the RECORD checklist with your revised ms. Please ensure that this is available as a supplementary file ("S1_RECORD_Checklist" or similar) with your next revision; in the checklist, individual items should be referred to by section (e.g., "Methods") and paragraph number rather than by line or page numbers, as the latter generally change in the event of publication. 

*** Reviewer 4 comments:

No further comments

*** Reviewer 5 comments:

The authors have extensively revised and much  improved the manuscript in accordance with the reviewers' numerous comments and suggestion. I must complement them on their thorough consideration of all the comments.

I have only one small suggestion, which the authors can feel free to adjust to or to discard, about their response to my comment 5 Methods, page 4: 

The statement "The Swedish National Board of Health and Welfare does not recommend a countrywide population-based PSA screening program for men" is probably better supported by the following reference than by the dissertation.

Screening för prostatacancer. Rekommendation och bedömningsunderlag. Socialstyrelsen 2018. Access at https://www.socialstyrelsen.se/globalassets/sharepoint-dokument/artikelkatalog/nationella-screeningprogram/2018-10-15.pdf

***

[LINK]

---

## [Editor Report · Decision Letter 4]

8 Apr 2021

Dear Dr Fallah, 

On behalf of my colleagues and the Academic Editor, Dr Pinsky I am pleased to inform you that we have agreed to publish your manuscript "Risk of prostate cancer in relatives of prostate cancer patients  in Sweden: A nationwide cohort study" (PMEDICINE-D-20-02475R4) in PLOS Medicine. We apologize for the delay in sending you a decision. 

Prior to final acceptance, please: amend the text at the end of p.15 of the PDF to "... the rather high homogeneity of the Swedish population" (or similar); and remove the information on funding and competing interests at the end of the main text (this will appear in the article metadata via entries in the submission form). 

PRESS

Sincerely, 

Richard Turner, PhD 

rturner@plos.org